# Two independent modes of kidney stone suppression achieved by AIM/CD5L and KIM-1

Kyohei Matsuura[1,9], Natsumi Maehara[1,2,9], Aika Hirota[1,2], Ayaka Eguchi[1], Keisuke Yasuda[1,2], Kaori Taniguchi[1], Akemi Nishijima[1,2], Nobuyuki Matsuhashi[3], Yoshiyuki Shiga[4], Rumi Ishii[5], Yasuhiro Iguchi[6], Kazunari Tanabe[5], Satoko Arai [1,2] & Toru Miyazaki [1,2,7,8✉]

The prevalence of kidney stones is increasing and its recurrence rate within the first 5 years is over 50%. No treatments that prevent the occurrence/recurrence of stones have reached the clinic. Here, we show that AIM (also called CD5L) suppresses stone development and improves stone-associated physical damages. The N-terminal domain of AIM associates with calcium oxalate crystals via charge-based interaction to impede the development of stones, whereas the 2nd and C-terminal domains capture the inflammatory DAMPs to promote their phagocytic removal. Accordingly, when stones were induced by glyoxylate in mice, recombinant AIM (rAIM) injection dramatically reduced stone development. Expression of injury molecules and inflammatory cytokines in the kidney and overall renal dysfunction were abrogated by rAIM. Among various negatively charged substances, rAIM was most effective in stone prevention due to its high binding affinity to crystals. Furthermore, only AIM was effective in improving the physical complaints including bodyweight-loss through its DAMPs removal effect. We also found that tubular KIM-1 may remove developed stones. Our results could be the basis for the development of a comprehensive therapy against kidney stone disease.

[1] Laboratory of Molecular Biomedicine for Pathogenesis, Center for Disease Biology and Integrative Medicine, Faculty of Medicine, The University of Tokyo, Tokyo 113-0033, Japan. [2] The Institute for AIM Medicine, Tokyo 162-8666, Japan. [3] Department of Gastroenterology, NTT Medical Center Tokyo, Tokyo 141-8625, Japan. [4] Department of Urology, Robotic Surgical Center, NTT Medical Center Tokyo, Tokyo 141-8625, Japan. [5] Department of Urology, Tokyo Women's Medical University, Tokyo 162-8666, Japan. [6] Medical corporation JISEIKAI, Tokyo 132-0003, Japan. [7] LEAP, Japan Agency for Medical Research and Development, Tokyo 113-0033, Japan. [8] Laboratoire d'ImmunoRhumatologie Moléculaire, Plateforme GENOMAX, Institut National de la Santé et de la Recherche Médicale UMR_S 1109, Faculté de Médecine, Fédération Hospitalo-Universitaire OMICARE, Fédération de Médecine Translationnelle de Strasbourg, Laboratory of Excellence TRANSPLANTEX, Université de Strasbourg, Strasbourg, France. [9]These authors contributed equally: Kyohei Matsuura, Natsumi Maehara. ✉email: tm@iamaim.jp

Kidney stone formation is highly prevalent and has been increasing with recurrence rate up to 50% within the first 5 years after the initial stone episode in both sexes over the past 50 years, owing to rapid changes in lifestyle and dietary habits as well as global warming[1–6]. Metabolic syndrome associated with obesity, diabetes, and hypertension is considered a strong risk factor for stone formation. Conversely, stone formers are at risk of hypertension, acute kidney injury (AKI), and chronic kidney disease[6–16]. Globally, ~80% of kidney stones are composed of calcium oxalate (CaOx) mixed with calcium phosphate, 10% of struvite, 9% of uric acid, and the rest are composed of cystine or ammonium acid urate or are diagnosed as drug-related stones[17,18].

The excessive supersaturation in urine results in crystal nucleation, the initial step in the transformation from a liquid to a solid phase within the urinary tract[19–22]. CaOx crystals attach to the surface of renal tubular epithelial cells by interacting with negatively charged membrane components such as phosphatidylserine, which is redistributed to the cell surface upon mechanical (by the crystals themselves) and/or chemical (by oxalate) cell injury[23,24]. When the attached crystals are retained at the luminal side of tubular epithelial cells, they aggregate, grow, and develop into stones in the urinary tract (nephrolithiasis or urolithiasis)[25–31]. Some crystals are endocytosed by epithelial cells, and thereafter, dissolved within lysosomes, or re-emerge at the basolateral surface, again providing centers for stone growth in the renal interstitial area[32–35]. In addition, crystal uptake often damages cells and causes epithelial cell death, which releases cellular debris that forms a nidus of additional crystal growth, thereby promoting stone formation[36,37]. Thus, it appears that other than urinary supersaturation, crystal-cell attachment, crystal aggregation/growth, and tubular epithelial cell damage are highly important processes during stone formation. Although tremendous efforts have been directed toward the development of effective therapies against kidney stone disease targeting either or all of these processes, no medical treatments that effectively prevent the occurrence/recurrence of kidney stones or induce stone removal have reached the clinic, other than behavioral and nutritional interventions or surgical/endourological treatment. Consequently, the annual cost of treatment for kidney stones and associated complaints is currently over 2 billion dollars and increasing in the United States alone[1,38].

Apoptosis inhibitor of macrophage (AIM; also called CD5 antigen-like: CD5L) is a circulating protein produced by tissue macrophages, which we initially identified as a supporter of macrophage survival[39] and is now known as a facilitator of repair in many diseases[40–45]. Of the mechanisms promoting disease repair by AIM, the enhanced phagocytic removal of dead cell debris and dead cell-derived damage-associated molecular patterns (DAMPs) has been most highlighted. We demonstrated that AIM associates with cell debris or DAMPs via charge-based interactions using a unique positively charged amino-acid cluster at its carboxyl terminus within the third scavenger receptor cysteine-rich (SRCR) domain[39], as well as disulfide bond formation using the solitary cysteine residue located at the second SRCR domain[46,47]. This association strongly enhances the engulfment of debris and DAMPs by phagocytes because AIM is highly internalized by phagocytes via multiple scavenger receptors[45,47]. Indeed, through such actions, the intravenous injection of recombinant AIM (rAIM) protein promotes the repair of ischemia/reperfusion-induced AKI by improving the obstruction of renal tubules and the associated sterile inflammation of the kidney[43,48], which are the central pathologies of AKI[49–51]. A similar therapeutic effect of AIM based on a comparable mechanism was confirmed recently in peritonitis and cerebral stroke in mice, in which the prognosis in disease animals

was markedly improved by rAIM administration[44,47]. In the present study, pathologic similarities between AKI and kidney stone disease, namely, the development of contaminants within renal tubules and subsequent luminal stricture/obstruction and sterile inflammation caused by DAMPs, led us to assess the possible impact of AIM as a therapeutic tool for kidney stone disease.

## Results

**AIM suppresses kidney stone development and associated tissue injury.** Repetitive glyoxylate administration is commonly employed to induce CaOx-based kidney stones in animals. The amount of stones increases mainly at the cortico-medullary junction, where the proximal renal tubules are located, over a period of 1 week of administration in mice[52].

Impressively, when rAIM (400 μg) was injected intravenously in mice on days 1, 3 and 5 during glyoxylate administration, the amount of kidney stones at day 6 was dramatically decreased (Fig. 1a; the SDS-PAGE result of the rAIM is presented in Supplementary Fig. 1a). Quantitative PCR analysis revealed that the mRNA levels of NGAL and KIM-1, typical markers of tubular injury, were lower in rAIM-injected mice than in non-injected mice, suggesting that stone-associated epithelial cell injury was reduced by rAIM (Fig. 1b). In addition, the mRNA levels of inflammatory cytokines such as IL-1β, IL-6, TNFα, and MCP-1 as well as myeloid cell markers CD11b and F4/80 at day 6 were also reduced significantly by rAIM injection, demonstrating the suppression of sterile inflammation at the lesion (Fig. 1c). The serum levels of creatinine (Cre) and blood urea nitrogen (BUN) were also decreased upon rAIM injection, indicating that AIM improved the kidney damage caused by stone-associated tissue injury and inflammation (Fig. 1d). Glyoxylate challenge decreased the bodyweight of the mice and deteriorated their general physical state, likely associated with kidney damage. rAIM injection also improved such pathological phenotypes including bodyweight loss (Fig. 1e) and decreased food intake (Fig. 1f).

KIM-1, which is highly expressed at the lumen of proximal tubules upon tubular epithelial cell injury, also acts as the counterpart of AIM in the phagocytic removal of tubular-obstructing debris during the repair process of AKI[43,48]. There, KIM-1 behaves as a scavenger receptor that promotes the engulfment of AIM-associated dead cell debris by tubular epithelial cells. Because KIM-1 mRNA levels increased by up to 600-fold upon glyoxylate loading (Fig. 1b), the hypothesis emerged that, similar to AKI, AIM might facilitate the phagocytic removal of CaOx crystals in collaboration with KIM-1, thereby reducing intraluminal stone development. Unexpectedly, however, when KIM-1-deficient (KIM-1−/−) mice[48] were injected intravenously with rAIM from day 1 of glyoxylate loading, the amount of kidney stones formed on day 6 was decreased to a level comparable to that observed in wild-type mice (Fig. 1g). Thus, unlike in AKI, it appears that the decrease of kidney stones by rAIM treatment was achieved independently of the AIM/KIM-1 axis.

**AIM associates with CaOx crystals and impedes their growth.** To address the mechanism by which AIM prevents the development of kidney stones in mice, we first asked whether AIM induces the phagocytic removal of CaOx crystals by tubular epithelial cells via certain scavenger receptors expressed on these cells to dissolve them in intracellular lysosomes. Indeed, we previously demonstrated that AIM is recognized by CD36[40], a well-known scavenger receptor expressed on proximal tubular epithelial cells[53]. Hence, we tested the uptake of CaOx crystals by mProx24 cells, a mouse proximal tubular epithelial cell line[43], in

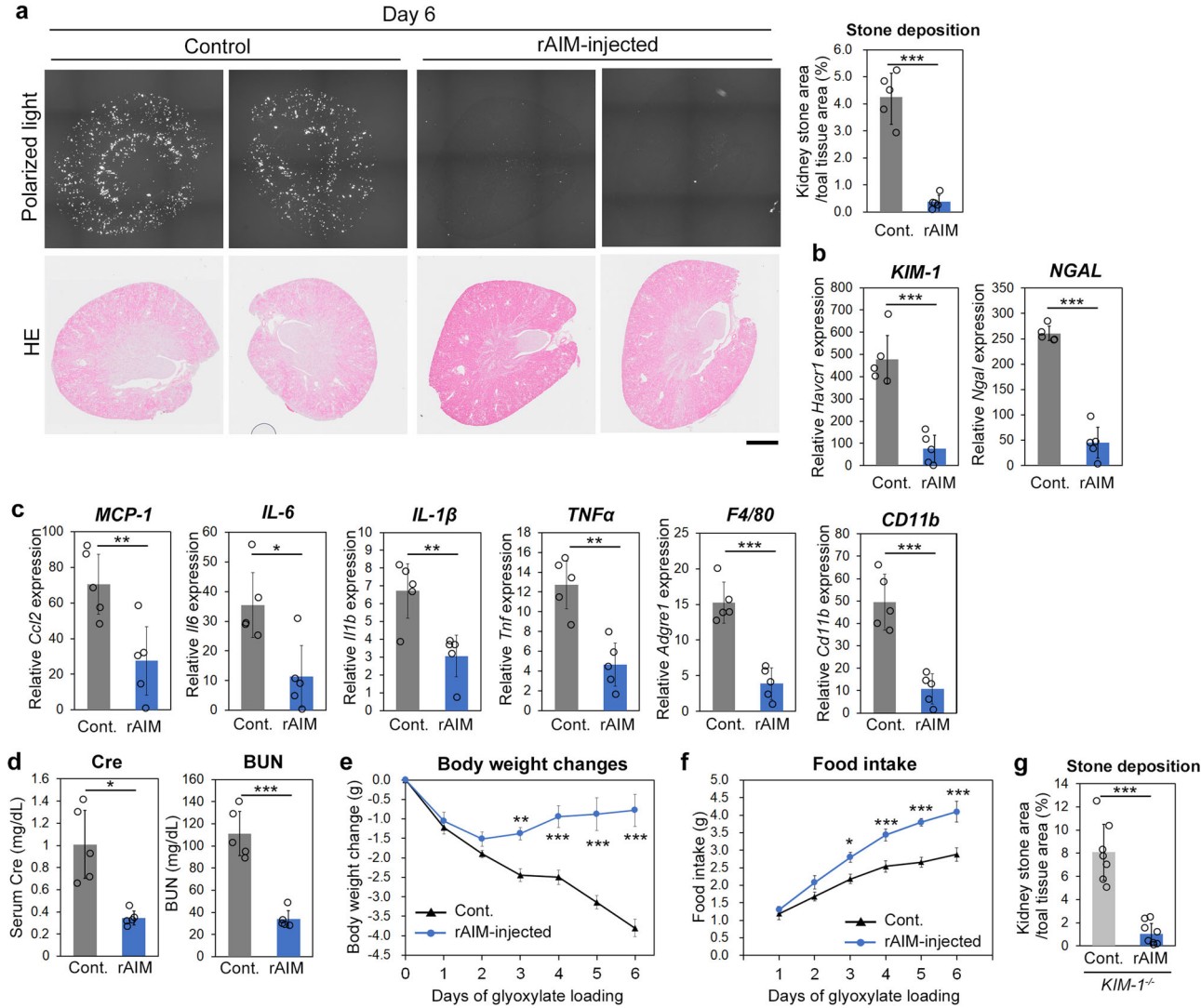

**Fig. 1 rAIM prevents kidney stone development. a** The amount of kidney stones in rAIM-injected (400 μg; $n = 5$) and PBS-injected ($n = 5$) mice on day 6 of glyoxylate loading. The number represents the percentage of stone area in the whole kidney section. Scale bar: 1 mm. The average stone area in three non-serial sections for each mouse is plotted. **b**, **c** Quantitative PCR analysis of the mRNA levels of kidney injury markers, *Haver1* (*KIM-1*) and *Ngal* (**b**), and various pro-inflammatory genes and macrophage marker genes (**c**) in the kidney of the mice above. **d** Serum Cre and BUN levels. **e** Bodyweight changes. Values are relative to those before glyoxylate loading. **f** Daily food intake. **g** The amount of kidney stones in rAIM-injected ($n = 7$) and PBS-injected ($n = 8$) *KIM-1*$^{-/-}$ mice on day 6 of glyoxylate loading. Means ± s.d. (**a–d**, **g**) or s.e.m (**e**, **f**) are shown. Statistical analysis was performed using Welch's t-test (**a–d**, **g**) or repeated measures ANOVA (**e**, **f**).

the presence or absence of rAIM in vitro. Flow cytometric analysis demonstrated that the presence of rAIM did not influence (or even disturb) the capture and/or incorporation of CaOx crystals by mProx cells (Supplementary Fig. 2), excluding the possibility of the enhancement of phagocytic CaOx crystal removal by AIM.

We then thought that AIM might disturb the aggregation and growth of CaOx crystals. Therefore, we incubated an equivalent volume of 1 mM $CaCl_2$ and 5 mM $Na_2C_2O_4$ in the presence or absence of rAIM and observed the development of CaOx crystals under a microscope. As expected, the number and size of crystals were apparently smaller in the presence of rAIM. Moreover, the typical spiny shape of CaOx crystals was abolished and changed to a cobble-like round shape in the presence of rAIM (Fig. 2a).

Thus, AIM may associate with small CaOx crystals and impede their aggregation and growth. Indeed, the flowcytometric analysis clearly demonstrated the association of AIM and CaOx crystals (Fig. 2b). The binding of AIM to CaOx crystals was also

confirmed biochemically through a pull-down assay using FLAG-tagged rAIM (Fig. 2c; whole blot is presented in Supplementary Fig. 3). The SDS-PAGE result of the FLAG-tagged rAIM and rSRCRs are presented in Supplementary Fig. 1a and b. Further dissection of the binding mode of AIM to CaOx crystals using the three SRCR domains of AIM revealed that the N-terminal domain (SRCR1) efficiently associated with CaOx crystals, whereas the SRCR2 and SRCR3 domains did not show such obvious binding when tested biochemically (Fig. 2c). The surface of SRCR1 is highly negatively charged, while SRCR2 is neutral and SRCR3 is positively charged[46], similar to the calculated isoelectric point of each SRCR domain, which is 4.32, 8.44, and 6.55, respectively (Fig. 2c). Because CaOx crystals are positively charged due to the presence of calcium as a bridging cation[23], it is likely that SRCR1 binds to CaOx crystals via charge-based interactions, which impeded the aggregation and growth of small crystals. In accordance with this hypothesis, the SRCR1 domain, but not the SRCR2 or SRCR3 domain, effectively inhibited CaOx

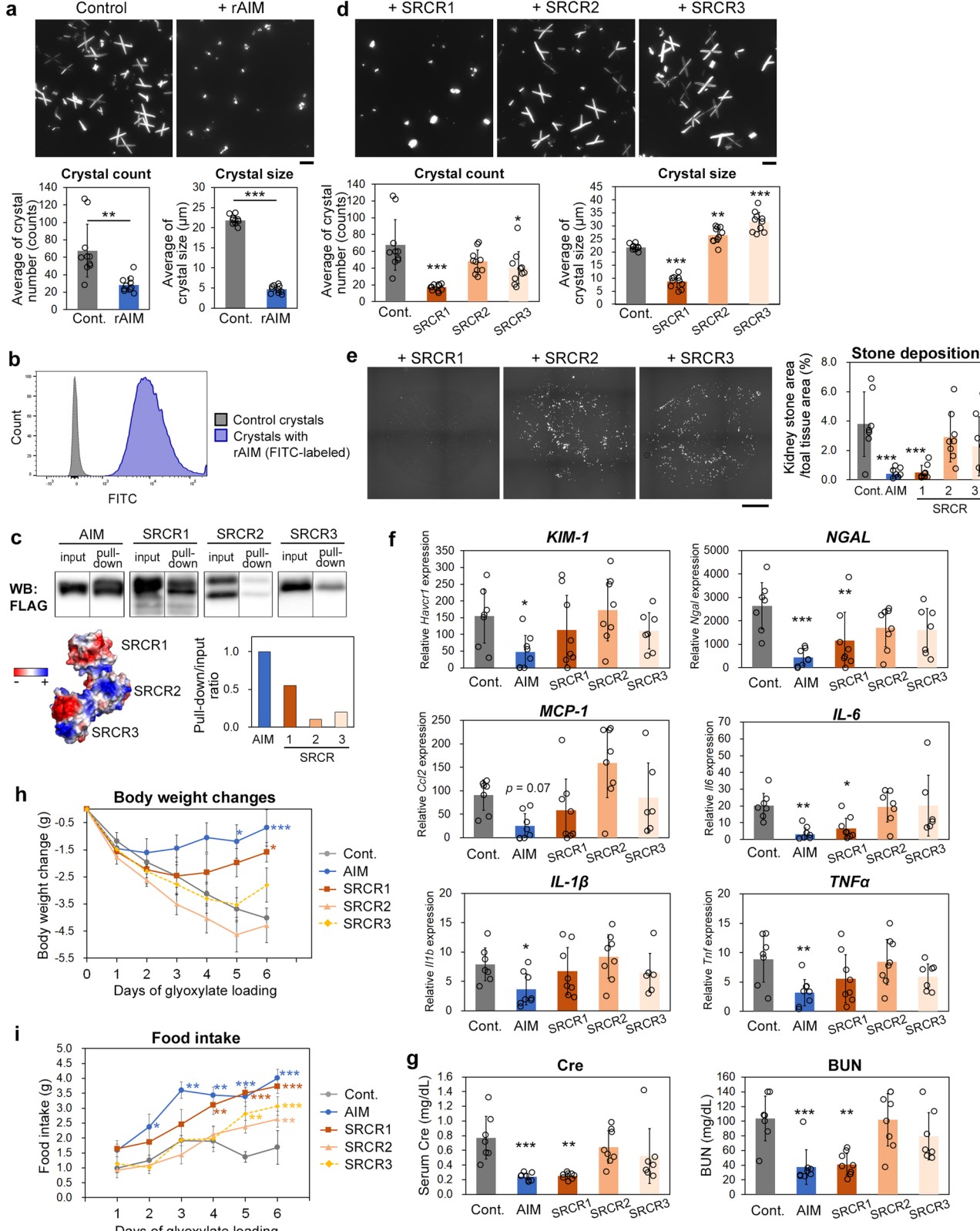

crystal growth in vitro (Fig. 2d). Moreover, intravenous injection of the SRCR1 domain at the equivalent molar level with that of rAIM decreased glyoxylate-induced stone development in mice (Fig. 2e). The mRNA levels of injury markers and inflammatory cytokines were reduced in the kidney (Fig. 2f), while serum Cre/BUN levels (Fig. 2g), bodyweight (Fig. 2h), and food intake (Fig. 2i) were also improved by SRCR1 administration.

Immunohistochemistry (IHC) for KIM-1 in the kidney specimens revealed that treatment with rAIM or SRCR-1 reduced the KIM-1 levels, whereas SRCR2 or SRCR3 did not obviously (Supplementary Fig. 4a). The ELISA analysis of mice sera for IL-6 showed the similar profile with that of QPCR result (Supplementary Fig. 4b). These effects were not observed by treatment with either the SRCR2 or SRCR3 domain (Fig. 2e–i). Thus, we conclude that

**Fig. 2 The mechanism by which AIM suppresses kidney stone development. a** Crystal development in the presence or absence of rAIM. Equivalent volume of $CaCl_2$ (2 mM) and $Na_2C_2O_4$ (10 mM) were mixed and incubated at 4 °C for 1 h in the presence or absence of rAIM (100 μg/mL). Representative photographs of the resulting crystals as well as the averages (± s.d.) of the number (/photo) and size (longest diagonal) of each crystal are presented. Ten photos taken randomly for each group were analyzed. Similar results were obtained in three independent experiments. Scale bar: 20 μm. **b** CaOx crystals were incubated with FITC-labeled rAIM (100 μg/mL) at 37 °C for 1 h, washed with PBS, and their association was analyzed using a flow cytometer. Similar results were obtained in three independent experiments. A representative result is shown. **c** Pull-down assay. CaOx crystals were incubated with either rAIM, SRCR1, SRCR2, or SRCR3 domain (FLAG-tagged; 1 μg/mL for rAIM and 0.3 μg/mL for each SRCR domain) at 37 °C for 1 h, washed with PBS, and precipitated by centrifugation. The association of FLAG-tagged rAIM and SRCR domains with CaOx crystals was assessed by immunoblotting using an anti-FLAG antibody after the precipitated crystals were boiled in an SDS-PAGE-loading buffer. The extra bands for the FLAG-tagged SRCR1 and SRCR2 domains are likely to be derived from differences in glycosylation. The white line in the middle of the band of pull-downed rAIM-FLAG is a halation due to the too high intensity of the signal and does not represent two bands. Ratio of pull-down/input signals is presented. A scheme of AIM with the charge-distribution[46] is presented. Blue: positively charged amino acid; red: negatively charged amino acid; and white: neutral amino acid. **d** In vitro crystal development carried out as in **a** in the presence of each FLAG-tagged SRCR domain. Similar results were obtained in three independent experiments. Scale bar: 20 μm. **e** Kidney stone development in mice at day 6 of glyoxylate loading with injection of FLAG-tagged SRCR domain (120 μg for each) on days 1, 3, and 5. Data are presented as in Fig. 1a. 7–8 mice for each group were analyzed. Scale bar: 1 mm. **f** Quantitative PCR analysis of the mRNA levels of kidney injury markers and various pro-inflammatory genes in the kidney of the mice above. **g** Serum Cre and BUN levels. **h** Bodyweight changes. Values are relative to those before glyoxylate loading. **i** Daily food intake. Means ± s.d. (**a**, **d–g**) or s.e.m. (**h**, **i**) are shown. Statistical analysis was performed with Welch's $t$-test (**a**), or one-way (**d–g**) or repeated measures (**h**, **i**) ANOVA followed by Dunnett's post hoc test. Significance is added when there is a significant difference against the control.

AIM prevented kidney stone development by binding to small CaOx crystals via SRCR1 to disrupt their aggregation and growth. It may be noteworthy that although the reduction of stones and the improvement in serum Cre/BUN levels were achieved comparably by the whole AIM protein and the SRCR1 domain, the suppression of kidney inflammation and the improvement of bodyweight/food intake were more prominent when the mice were injected with the whole AIM protein compared with only the SRCR1 domain.

Intriguingly, despite the apparent preventive effect of AIM on kidney stone development, when stones were induced in AIM-deficient ($AIM^{-/-}$) mice, the amount of stones on day 6 was comparable to that in wild-type mice (Supplementary Fig. 5a). This was in contrast to the case for AKI, where the disease progresses more profoundly in $AIM^{-/-}$ mice compared with wild-type mice[43,48]. In both mice and humans, serum AIM is usually present in association with pentameric IgM and is released during AKI to facilitate disease repair[43,46,48]. We observed less induction of IgM-free AIM in serum in mice with kidney stones compared with those with AKI (Supplementary Fig. 5b). In human patients with kidney stones, no significant increase was observed in serum IgM-free AIM levels compared to those in healthy individuals (Supplementary Fig. 5c). In addition, immunohistochemistry in mice revealed that there was no obvious AIM staining on intraluminal kidney stones (Supplementary Fig. 5d), which was contrastive to the massive accumulation of AIM at the intratubular dead cell debris during AKI[43]. Thus, although AIM protein is potent for the suppression of kidney stone development, it appeared that the presence of crystals did not induce the release of sufficient endogenous AIM from IgM pentamer in blood in mice and humans.

**Negatively charged substances suppress kidney stone development at different levels**. Having identified the unique mechanism by which AIM prevents kidney stone development, we next asked whether different negatively charged substances could also inhibit CaOx crystal growth and the development of kidney stones. To this end, we tested recombinant osteopontin (rOPN) protein, acidic polyanion poly(acrylic acid) (5.1 kDa; $pAA_{5.1}$), and a highly acidic artificial peptide (DEDDDEDDDEDD; $D_9E_3$) for their therapeutic effect in comparison with that of rAIM. OPN is a highly negatively charged extracellular matrix protein that exhibits potent in vitro inhibitory activity toward calcium crystal aggregation and adhesion to renal epithelial cells[54–57]. However,

because a possible role for OPN as a promoter of stone formation by tethering CaOx crystals to tubular cell membranes has also been proposed, the mode of involvement of OPN in the process of kidney stone development remains controversial[58–60]. Continuous infusion of the acidic polyanion $pAA_{5.1}$ has been shown to suppress CaOx crystal deposition in rat kidney[61,62], although the molar amount of $pAA_{5.1}$ per bodyweight given daily was 10–100 times larger than that of rAIM used in the present study.

We first confirmed the inhibitory effect of the substances on the aggregation and growth of CaOx crystals in vitro. We also used a highly positively charged artificial peptide (RRRRKRKRKRKR; $R_8K_4$) as a control of the $D_9E_3$ peptide. All negatively charged substances tested including rAIM added at an equivalent molar level decreased the size and spiny shape of CaOx crystals (Fig. 3a). Interestingly, their effects on the overall number of developed crystals were variable; the presence of rOPN even increased the number of crystals (Fig. 3a). As expected, the $R_8K_4$ peptide had no effect (Fig. 3a). We then addressed their therapeutic effect against kidney stone development and associated pathological symptoms in vivo. During glyoxylate administration to mice for 6 days, each substance was administered intravenously (rOPN and $D_9E_3$ peptide) or intraperitoneally ($pAA_{5.1}$) at the equivalent molar amount with 400 μg rAIM on days 1, 3, and 5. All three substances reduced the resulting amount of kidney stones at day 6 of glyoxylate treatment, but not as effectively as rAIM (Fig. 3b). Likewise, the reduction of serum Cre/BUN levels was not as prominent as that achieved by rAIM treatment (Fig. 3c). It was also the case for the decrease in the levels of injury markers and inflammatory molecules (Fig. 3d and Supplementary Fig. 6). More impressively, the recovery of body-weight observed with rAIM was not induced by the three substances (Fig. 3e). Similarly, neither negatively charged substance improved food intake (Fig. 3f). Overall, these results suggest the prominent efficacy of rAIM as a therapeutic tool against kidney stone development and the associated deterioration of the general physical state. It is noteworthy that some (2~3 out of 7) mice died of acute renal failure due to massive stone development in control (PBS injected) or $R_8K_4$ peptide injected group, but none died in other groups. Intriguingly, in mice treated with $R_8K_4$ peptide, although stone development was not suppressed, mRNA levels for $IL$-$1β$ and $IL$-$6$ reduced significantly. While the exact reason for the effect is not clear, $R_8K_4$ peptide might possess a direct anti-inflammatory effect like the Tryptophan-Histidine peptide, which is also positively charged and was reported to be anti-inflammative through blockade of L-type $Ca^{2+}$ channels[63].

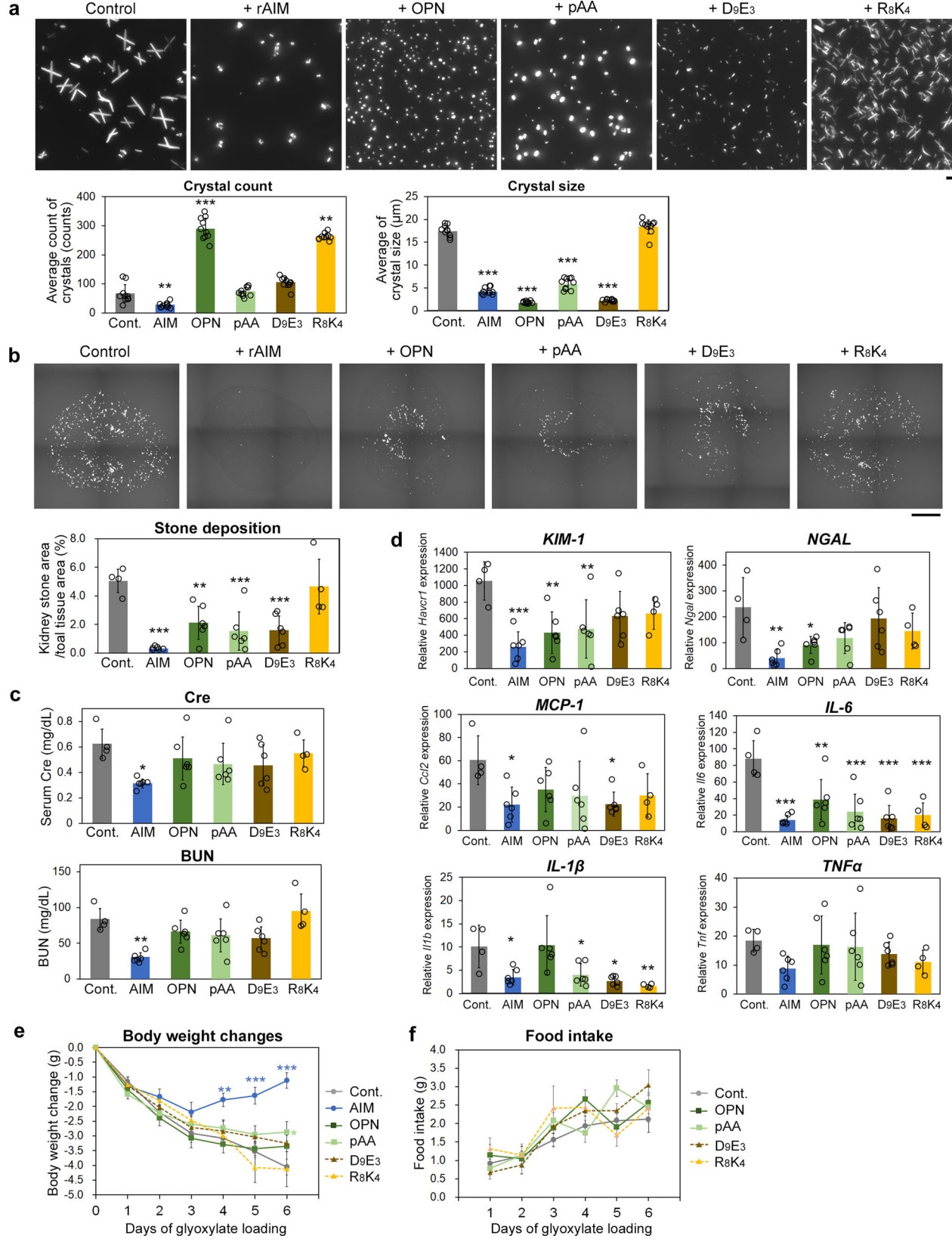

**AIM associates with crystals more strongly than other negatively charged substances**. As a reason for the predominant effect of rAIM in preventing kidney stone development among different negatively charged substances, AIM might attach to CaOx crystals more covalently than the other compounds to impede stone development more effectively. We tested this idea by using a competitive binding assay. CaOx crystals were incubated with rAIM and either rOPN, $pAA_{5.1}$, or $D_9E3$ peptide at different molar ratios, and the interference of rAIM binding to CaOx crystals by the presence of the other substances was assessed biochemically. The presence of any substance at an equivalent molar level did not disturb the association of rAIM to CaOx

**Fig. 3 Therapeutic efficacy of rAIM and different negatively charged substances against kidney stones and associated deterioration of the general physical state. a** In vitro crystal development carried out as in Fig. 2a in the presence of either rAIM, rOPN, $pAA_{5.1}$, $D_9E_3$ peptide, or $R_8K_4$ peptide at identical molar levels (rAIM; 100 µg/mL, rOPN; 125 µg/mL, $pAA_{5.1}$; 15 µg/mL, $D_9E_3$ and $R_8K_4$ peptide; 2.5 µg/mL each). Representative photographs of the resulting crystals as well as the averages (± s.d.) of the number (/photo) and size (longest diagonal) of each crystal are presented. Similar results were obtained in three independent experiments. Scale bar: 20 µm. **b** Kidney stone development in mice at day 6 of glyoxylate loading with injection of the indicated substance at identical molar levels (rAIM; 400 µg, rOPN; 500 µg, $pAA_{5.1}$; 60 µg, $D_9E_3$ and $R_8K_4$; 10 µg each) on days 1, 3 and 5. Data are presented as in Fig. 1a. $n = 4$ (control and $R_8K_4$ peptide), 6 (rAIM, OPN, pAA and $D_9E_3$ peptide). Note 3 control mice and 3 $R_8K_4$-injected mice (out of 7 mice at the beginning of the experiment) died of kidney failure before day 6. Scale bar: 1 mm. **c** Serum Cre and BUN levels. **d** Quantitative PCR analysis of the mRNA levels of kidney injury markers and various pro-inflammatory genes in the kidney of the mice above. **e** Bodyweight changes. Values are relative to those before glyoxylate loading. **f** Daily food intake (without rAIM group). Means ± s.d. (**a–d**) or s.e.m. (**e**, **f**) are shown. Statistical analysis was performed with one-way (**a–d**) or repeated measures (**e**, **f**) ANOVA followed by Dunnett's post hoc test. Significance is added when there is a significant difference against the control.

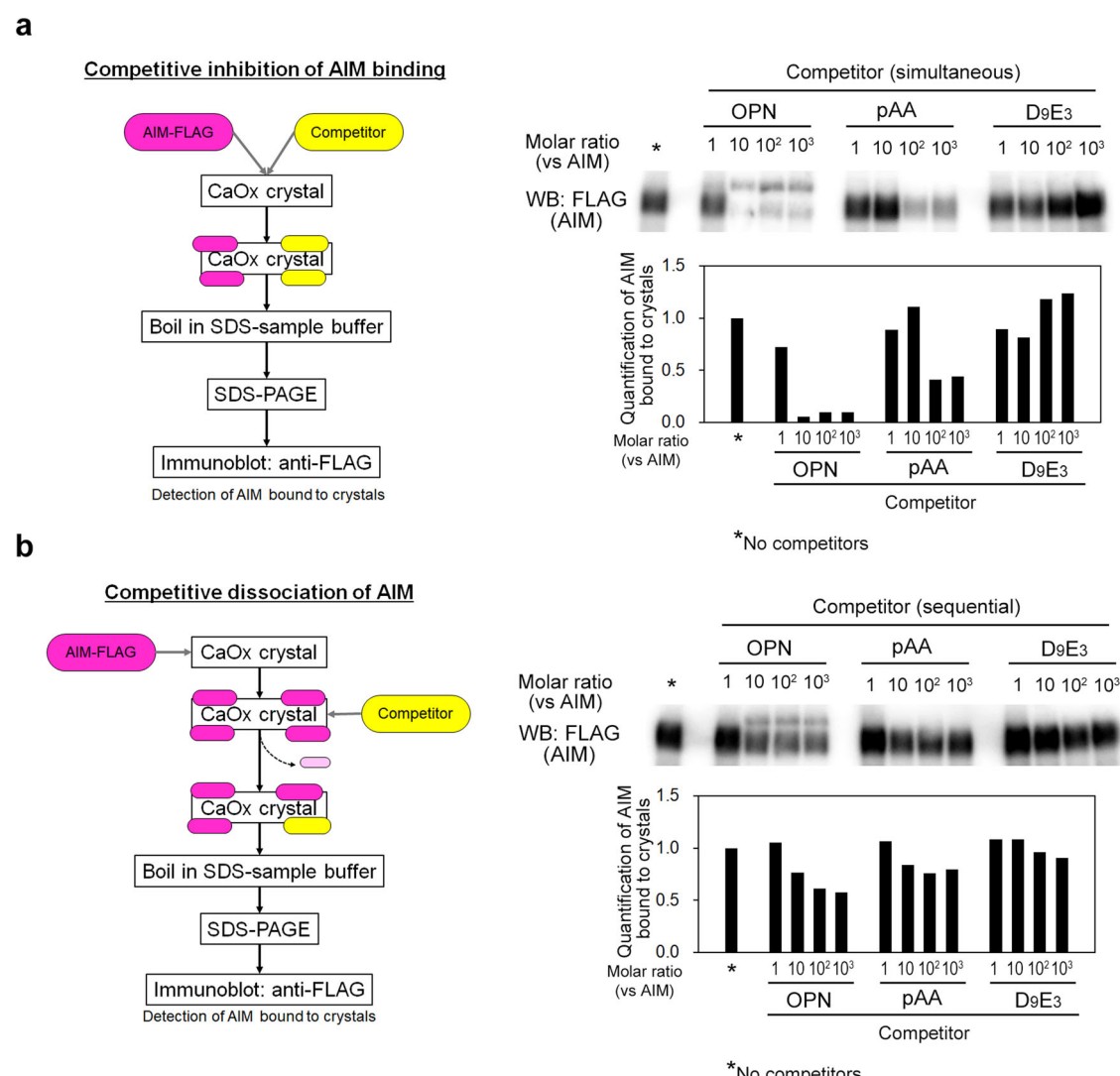

**Fig. 4 High binding affinity of rAIM to crystals. a** Crystals were incubated with FLAG-tagged rAIM (0.5 µg/mL) and either rOPN, $pAA_{5.1}$, $D_9E_3$ peptide (presented as competitor), or PBS (presented as *) at different molar ratios at 37 °C for 1 h. Thereafter, the crystals were washed with PBS, boiled in an SDS-PAGE loading buffer, and the rAIM bound to the crystals was assessed by immunoblotting. The values of AIM signals are relative to that of the crystals incubated with only rAIM (left lane indicated as *). **b** Crystals were pre-incubated with rAIM (0.5 µg/mL) at 37 °C for 1 h, and thereafter washed with PBS. The crystals were then incubated with the indicated substances at different molar ratios relative to that of rAIM. After additional incubation for 1 h, the remaining rAIM on the crystals was assessed as in **a**. In both experiments, representative results are presented. Similar results were obtained in three independent experiments.

crystals (Fig. 4a; the whole blot is presented in Supplementary Fig. 7a). This tendency was maintained when $pAA_{5.1}$ or D9E3 peptide was added at 100- or 1000-fold higher molar levels (Fig. 4a). In addition, rAIM that had been pre-bound to CaOx crystals was not released when the crystals were incubated with the other substances at identical or even higher molar levels (Fig. 4b; the whole blot is presented in Supplementary Fig. 7b). These results suggest that rAIM had superior binding affinity to CaOx crystals compared with the other negatively charged substances. In addition, intravenously injected rOPN might not be

excreted to urine efficiently to reach intraluminal crystals because it is larger than rAIM (40 kDa AIM vs. 45–66 kDa OPN, depending on glycosylation levels). Indeed, no urinary rOPN was detected (only a small amount of degraded C-terminal fragment was detected) (Supplementary Fig. 8), which was sharply contrastive to the rapid and efficient excretion of full-size rAIM after the injection[43].

**AIM reduces DAMPs in stone-containing kidney**. Among the negatively charged substances tested in this study, only rAIM improved the bodyweight and food intake of glyoxylate-loaded mice (Fig. 3e, f). We recently reported that AIM binds to DAMPs and promotes their removal by phagocytes, which markedly improves the physical states of animals with cerebral infarction and their overall prognosis[47]. When CaOx crystals accumulate in the kidney, considerable amounts of DAMPs must be released from injured/dead cells at the lumen and interstitial regions of the kidney, thereby promoting renal inflammation. Therefore, removal of DAMPs by rAIM may ameliorate the physical complaints associated with kidney stone. It is not expected that rOPN, $pAA_{5.1}$, or D9E3 peptide would possess such potency because they do not have solitary cysteine residues or positively charged amino acid clusters, both of which are required for effective binding to DAMPs.

To this end, we performed immunohistochemistry for S100A9, one of the most representative DAMPs in the kidney[64–66], using mouse kidney specimens from day 6 of glyoxylate loading and quantified the area of extracellular S100A9 staining (that did not overlap with DAPI-stained nuclei). As expected, treatment with rAIM decreased the volume of S100A9 staining in the lumen and interstitial area of the kidney most remarkably (Fig. 5). Note that the area of extracellular S100A9 staining was also reduced at different levels in mice injected with rOPN, $pAA_{5.1}$, or D9E3 peptide, but not to the level achieved by rAIM injection (Fig. 5).

**KIM-1 mediates the removal of developed stones in an AIM-independent fashion**. Upon repetitive injection of glyoxylate in mice, the number of kidney stones increased during a certain period (~6 days); however, interestingly, it tended to decrease afterward, even under continuous glyoxylate loading (Fig. 6a and ref. [52]). This suggests that certain machinery to remove developed stones is induced upon stone accumulation in the kidney.

Although KIM-1 was not involved in the prevention of stone development by AIM, its expression was highly induced at the lumen during stone development and its function as a scavenger receptor led us to suppose that KIM-1 might play a role in the physiological removal of developed stones. Supporting this idea, when $KIM-1$ mRNA levels were analyzed sequentially during glyoxylate-induced stone formation, they increased markedly on days 3 and 6 when stone accumulation exhibited maximum levels, and thereafter decreased on day 9 when the number of stones was already considerably reduced compared with day 6 (Fig. 6b). In the histologic analysis, tubular epithelial cells surrounding the deposited stones were associated with KIM-1 staining, suggesting that tubular epithelial cells may remove stones actively though KIM-1 (Fig. 6c).

The vital role of KIM-1 in stone removal was corroborated by inducing kidney stones by glyoxylate in $KIM-1^{-/-}$ mice. The number of stones on days 3 and day 6 was significantly higher in $KIM-1^{-/-}$ mice compared with wild-type mice, and more importantly, on day 9, the number of stones was further increased in $KIM-1^{-/-}$ mice, in sharp contrast with their significant decrease in wild-type mice (Fig. 6d). All of these data indicated that the induction of KIM-1 in tubular epithelial cells is an important defense response against stone accumulation.

Interestingly, daily administration of rAIM into stone-harboring $KIM-1^{-/-}$ mice after day 6 of glyoxylate loading did not decrease the number of stones toward day 9 (Fig. 6e). In addition, rAIM administration to wild-type mice did not enhance the natural reduction of kidney stones (Fig. 6e). Thus, rAIM is highly potent in the prevention of kidney stone formation, but not for their removal once they have developed.

**Discussion**

The findings of this study suggest two distinct mechanisms that are potentially therapeutically applicable to kidney stone disease: the prevention of stone development by AIM and the removal of developed stones by KIM-1.

In particular, we showed that AIM is effective at preventing the occurrence of kidney stones. It is likely that AIM could decrease the recurrence of kidney stones, which is important and practical, considering the high and still increasing recurrence rate globally and the fact that no effective treatments preventing the recurrence of kidney stones have been developed. As for the underlying mechanism, we were surprised to find that binding of the SRCR1 domain of AIM to CaOx crystals prevented stone development by interfering with crystal aggregation and growth. This is a unique action of AIM that is entirely different from its behavior in other diseases[43,44,47,48]. In most cases, AIM associates with the pathological targets such as dead cell debris and DAMPs via SRCR2 and SRCR3 and binds to scavenger receptors on the surface of phagocytes through SRCR1, thereby promoting their phagocytic removal, leading to disease repair. In addition to the suppression of CaOx crystal growth, the association of AIM may also impede the attachment of crystals to phosphatidylserine on the surface of tubular epithelial cells by neutralizing the positive charge of the crystals. This action may further suppress stone development because the attachment of CaOx crystals to epithelial cells causes the retention of crystals at the lumen (via their surface thorns), cell injury, and the production of dead cell debris, which becomes the core of the stone.

Based on its strong binding affinity to CaOx crystals, AIM was most effective in suppressing kidney stone development among the various negatively charged substances tested in this study. Furthermore, AIM was the only material that also improved the physical complaints of mice associated with the stone development. This outcome is reminiscent of our previous observation that AIM reverses the massive bodyweight loss and improves the general physical state in ischemia/reperfusion-induced AKI in mice[43]. This was also true in severe, lethal AKI in mice induced by ischemia/reperfusion plus high-salt loading[48]. Here, we demonstrated a marked decrease of S100A9, a representative DAMP in the kidney, at lesions and concluded that this consequence was the major reason for the predominancy of AIM in improving the stone-associated physical complaints. Most likely, it was also the case in AKI. Indeed, a chronic increase of DAMPs in the kidney might cause inflammation not only locally but also throughout the body. Intravenously injected rAIM might remove DAMPs from multiple tissues and thereby improve the health state. In addition, the strong effect of AIM in tissue repair at the proximal tubules, which was also observed in AKI[43,48], may reduce the serum levels of several uremic toxins that are excreted from the proximal tubules. Although global analysis of circulating uremic toxins in the presence or absence of rAIM is required to support this idea, this effect would certainly contribute to the recovery of the general physical state. Certainly, the decrease of kidney stones itself reduces renal cell injury, and thus, may reduce the production of DAMPs, as evident by the reduction of S100A9 in the kidney at a certain level even by rOPN, $pAA_{5.1}$, or D9E3 peptide, which are impotent to remove DAMPs (Fig. 5). Similarly,

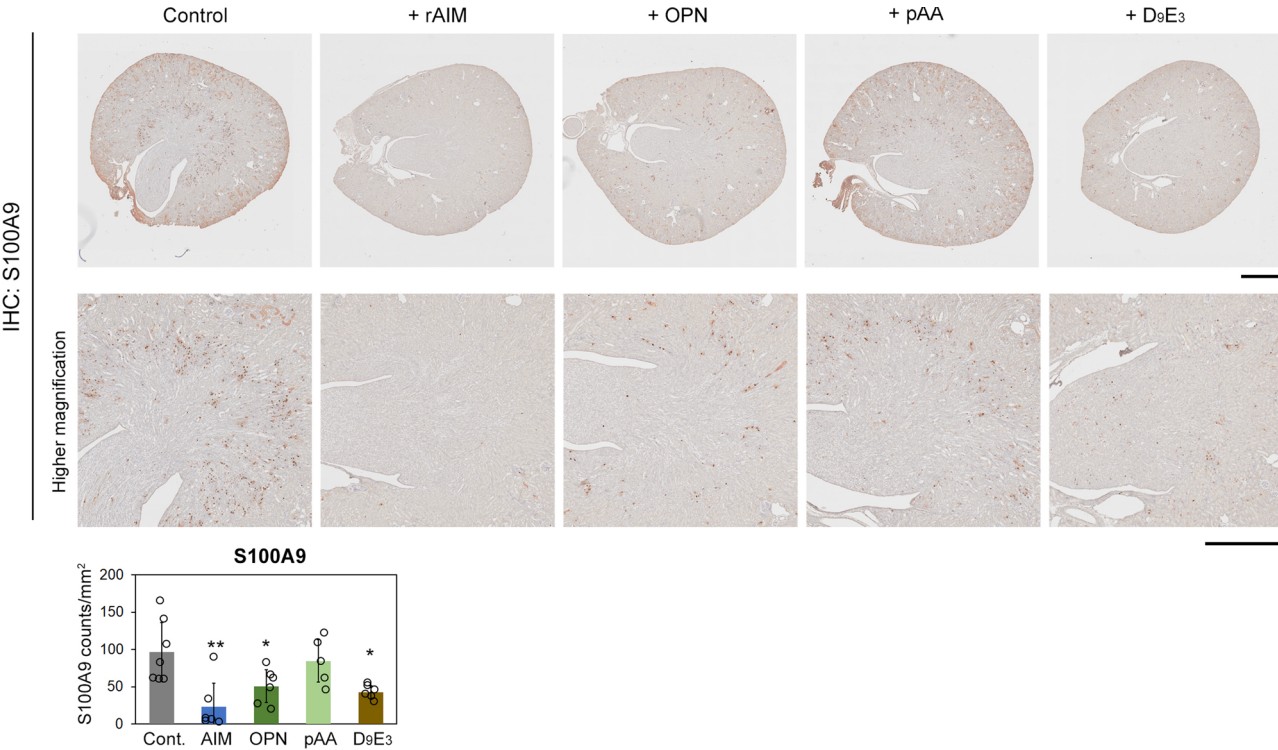

**Fig. 5 Removal of DAMPs in the kidney by rAIM.** Immunohistochemistry for S100A9 in kidney specimens from mice at day 6 of glyoxylate loading. Before analysis, the mice were injected with either rAIM, rOPN, pAA$_{5.1}$, D$_9$E$_3$ peptide, R$_8$K$_4$ peptide (amounts were as in Fig. 3b), or PBS at identical molar levels. Extracellular S100A9 positive area was analyzed. 7 control (PBS-injected) mice and 6 mice for other groups were analyzed. In pAA$_{5.1}$ injected mice, two circles are merged, and thus, one data point appears to be missing. Scare bars: 1 mm (for upper panels), 100 μm (for lower panels). Means ± s.d. are shown. Statistical analysis was performed with one-way ANOVA followed by Dunnett's post hoc test. Significance is added when there is a significant difference against the control.

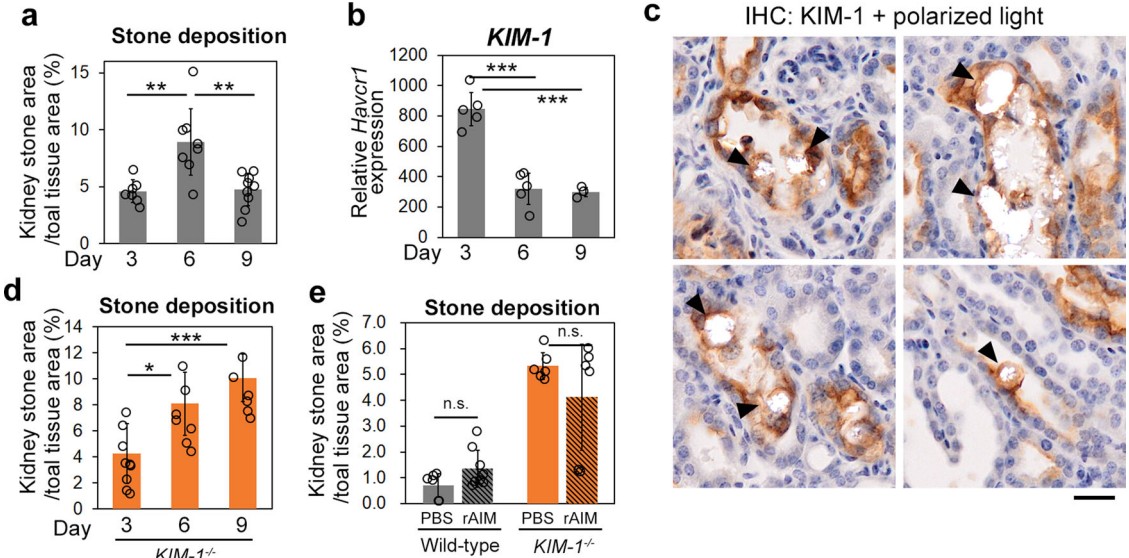

**Fig. 6 Removal of developed stones by KIM-1. a** Mice were injected with glyoxylate (150 mg/kg bodyweight) daily for 9 days and the amount of kidney stones was analyzed in mice on days 3, 6, and 9. $n = 7$–9. **b** The mRNA levels of Haver1 (KIM-1) in the kidney of the mice above. **c** Immunohistochemistry for KIM-1 on kidney specimens from mice on day 6 of glyoxylate loading. A representative photograph is presented. KIM-1 (brown) surrounding a stone is indicated. Scale bar: 20 μm. **d** The amount of kidney stones in KIM-1$^{-/-}$ mice on days 3, 6, and 9 of glyoxylate loading. $n = 6$–9. **e** Wild-type and KIM-1$^{-/-}$ mice were loaded with glyoxylate for 6 days to develop kidney stones, and thereafter injected daily with rAIM (400 μg) or PBS from day 6. On day 9, the mice were sacrificed and the amount of kidney stones was analyzed. $n = 6$–8. Scale bar: 20 μm. Means ± s.d. are shown. Statistical analysis was performed with one-way ANOVA followed by Bonferroni's post hoc test (**a**, **b**, **d**), or Welch's t-test in each genotype group (**e**).

the subtle recovery of bodyweight and food intake by the SRCR1 domain, which is not able to bind to and remove DAMPs directly, may be due to the strong suppressive effect of SRCR1 in kidney stone development. Thus, it is likely that the dual effects of the whole AIM protein, the strong suppression of kidney stone development and the removal of DAMPs, achieved an effective, comprehensive therapeutic effect against kidney stone disease.

The pathophysiology of the mouse model with glyoxylate loading used in this study is similar to that of human primary hyperoxaluria type I (PH1), a rare autosomal recessive disease caused by mutations in the alanine–glyoxylate amino-transferase gene (AGXT)[67]. AGT (or AGT1), the protein encoded by AGXT, is a key enzyme in glyoxylate detoxification. In patients, insufficient AGT activity in peroxisomes leads to the increased conversion of glyoxylate to oxalate[67,68]. Excessive renal excretion of oxalate causes the loss of renal function because of CaOx deposition. In addition to renal damage, CaOx deposition becomes widespread and life-threatening unless liver and kidney transplantation is performed. Unfortunately, patients often experience the recurrence of multiple CaOx stones even after transplantation[67,69,70]. Thus, it is likely that rAIM treatment could be used to prevent the occurrence and post-therapy recurrence of stones in patients with PH1. The improvement of the general physical state by rAIM observed in the mouse model may provide the expectation for a cure of widespread, life-threatening CaOx deposition in PH1 patients. In other words, it might be possible that the widespread deposition of CaOx crystals also occurs in our mouse model of glyoxylate loading, which could also be a reason for the deterioration of their general physical state. If so, the improvement in bodyweight and food intake in mice following the administration of the SRCR1 domain may be explainable. Assessment of the effect of rAIM in animal models of PH1, such as AGT-deficient mice[71], will provide further evidence for its suitability in the treatment of human PH1 patients.

We were also excited to find that KIM-1 appeared to contribute to the removal of developed stones. Accumulating evidence has highlighted the role of various tissue epithelial cells as "semi-professional" phagocytes for their potential clearance of dead cells. For instance, bronchial epithelial cells engulf apoptotic cells and secrete anti-inflammatory cytokines, thereby suppressing airway inflammation[72]. Sandahl et al. reported that mammary tissue homeostasis and future lactation in the post-partum mammary gland are influenced by epithelial cell-directed apoptotic cell clearance[73]. We and others also reported that tubular epithelial cells remove intraluminal dead cell debris during AKI. However, it was surprising to observe that tubular epithelial cells might be potent not only in engulfing large stones but also in digesting them. It is noteworthy that after day 6 of glyoxylate loading, the number of stones was decreased at the lumen and interstitial area, suggesting that the engulfed luminal stones were digested intracellularly and were not re-distributed to the interstitial area. Although we believe such observations are important, many questions remain unsolved, e.g., the mechanism by which KIM-1 alone (without help from AIM) captures large stones and mediates their effective phagocytosis, and the precise machinery by which engulfed stones are guided to lysosomes and their effective digestion. Additional extensive studies are certainly required to elucidate these points. Nevertheless, according to such a strong scavenging potency of KIM-1 against kidney stones, KIM-1 overexpression could be worth considering as a new therapeutic strategy given that no effective treatment for the noninvasive removal of kidney stones has yet been developed. It is noteworthy, however, that Humphreys et al. showed that sustained KIM-1 expression in kidney epithelial cells caused renal fibrosis in KIM-1-overexpressing transgenic mice[74]. Therefore, the level and timing of KIM-1 expression in the kidney will need to be regulated finely.

## Methods

**Mice.** All animal experiments were carried out in strict accordance with the recommendations in the Guide for the Care and Use of Laboratory Animals of the National Institutes of Health. All surgeries were performed under sodium pentobarbital anesthesia, and all efforts were made to minimize suffering. During the animal experiments, we strictly complied with the requirement of a humane endpoint. We carefully observed the post-IR mice and euthanized the individual mouse and used for analysis, if either significant decrease in renal function (serum creatinine level 3.0 or higher), difficulty in feeding/water intake, symptoms of agony, long-term appearance abnormalities with no signs of recovery, rapid weight loss, was shown. The protocol was approved by the Committee on the Ethics of Animal Experiments of the University of Tokyo (Permit Number: P15-126 and P21-001).

**Human subjects and ethics.** Serum samples of individuals with or without kidney stones were obtained from Tokyo Women's Medical University Hospital. For analysis of human subjects, informed consent in writing was obtained from donors of serum, and the study protocol conformed to the ethical guidelines of the 1975 Declaration of Helsinki as reflected in a priori approval by the Ethics Committees of the University of Tokyo for Medical Experiments (Permission Number: 2019358NI) and Tokyo Women's Medical University (Permission Number: 2020-0016).

**Induction and evaluation of kidney stones in mice.** To induce kidney stones in mouse kidney, glyoxylate was introduced by daily intra-abdominal injection. The injection was performed according to the weight of each mouse (150 mg/kg) with a clean 27-gauge needle. To evaluate kidney stones, kidney specimens were fixed in 4% paraformaldehyde, and embedded in paraffin. Four-micrometer-thick cross-sections were dewaxed, and then sealed in the usual manner. Microscope (IX83, Olympus) with a polarized light optical lens was used to observe the sections, and pictures were analyzed by software HALO (Indica Labs).

**Antibodies and reagents.** Antibodies and reagents used for histological experiments are as follows: Primary antibodies are: KIM-1 (MAB1817, R&D systems), AIM (rab2 rabbit polyclonal for IHC of mouse and human kidney specimens); #11 and 12 (for human free AIM ELISA) established in our laboratory, partly purchasable from Transgenic Inc.), S100A9 (AF2065, R&D systems, NE, USA). Secondary antibodies and related reagents are: G-Block (Genostaff, Tokyo, Japan) and HISTOFINE simple stain mouse MAX-PO (R, Rat, or G) (for nucleus; NICHIREI, Japan). Specimens were analysed using an inverted microscope: IX83 (Olympus) and a research slide scanner: SLIDEVIEW VS200 (Olympus).

**Purification of rAIM.** CHO-S cells were transfected with pcDNA3.1-mAIM plasmid and cultured in CD Forti CHO medium (Invitrogen, CA) for 3 days. rAIM was purified from culture supernatant using rat anti-mouse AIM monoclonal antibody conjugated Protein G sepharose (GE Healthcare Life Sciences, PA). Bound protein was eluted with 0.1 M Glycin-HCl, pH 3.0, and neutralized with 1 M Tris-HCl, pH 8.5. Protein was concentrated as necessary using Amicon Ultra filter concentrators (Millipore, MA), and stored at $-80\,°C$ in PBS. Endotoxin levels were measured by the chromogenic LAL endotoxin detection system (Genscript, NJ) following the manufacturer's protocols. Protein concentration was determined by the BCA (bicinchoninic acid) assay according to the manufacturer's protocol (Pierce, Rockford, IL).

**Preparation of SRCR fragments.** ExpiCHO-S cells were transfected with pFLAG5.1-SRCR plasmid by ExpiFectamine CHO Transfection Kit (Gibco) and cultured with shaking in ExpiCHO Expression Medium (Gibco) for 4 days. Each SRCR fragment was purified from culture supernatant using anti-FLAG M2 affinity gel (Sigma-Aldrich). Bound protein was eluted with 0.1 M Glycin-HCl, pH 3.5 and neutralized with 1 M Tris-HCl, pH 8.5. Protein was concentrated as necessary using Amicon Ultra filter concentrators (Millipore, MA), and stored at $-80\,°C$ in PBS. Endotoxin levels were measured by the Limulus Color KY Test Wako (FUJIFILM Wako) following the manufacturer's protocols. Protein concentration was determined by the BCA (bicinchoninic acid) assay according to the manufacturer's protocol (Pierce, Rockford, IL).

**Negatively charged substances.** Flag-tagged recombinant osteopontin (OPN) protein was generated as SRCR fragments. Polyacrylic acid was purchased from FUJIFILM Wako. $D_9E_3$ and $R_8K_4$ peptides were synthesized by Pepmic Co., Ltd. (Jiangsu, China).

**Serum biomarkers.** Serum Cre concentrations were measured using a Lab-Assay Creatinine Kit (Wako Pure Chemical Co., Ltd., Osaka, Japan). Serum BUN levels

were determined using the FUJI DRI-CHEM 4000 V analyzer system (FUJIFILM Co., Ltd., Tokyo, Japan).

**Histology**. *IHC for AIM*: Kidneys were fixed in 4% paraformaldehyde in PBS for 24 h and embedded in paraffin. 8 μm sections were immunostained with the rabbit anti-AIM polyclonal antibody (Rab2; available for human and mouse AIM), followed by incubation with HISTOFINE simple stain mouse MAX-PO (R) (NICHIREI, Japan) for 30 min. After being stained with diaminobenzidine tetrahydrochloride (DAB), sections were counter-stained with hematoxylin. To block non-specific binding, slides were incubated with G-Block (GB-01, Genostaff) at r. t. for 20 min before immunostained. *IHC for KIM-1*: Kidneys were fixed in 4% paraformaldehyde in PBS) for 24 h and embedded in paraffin. 8 μm sections were immunostained with the rat anti-KIM1817 monoclonal antibody (MAB1817, R&D systems), followed by incubation with HITOFINE simple stain mouse MAX-PO (Rat) (NICHIREI, Japan) for 30 min. After being stained with diaminobenzidine tetrahydrochloride (DAB), sections were counter-stained with hematoxylin. To block non-specific binding, slides were incubated with G-Block (GB-01, Genostaff) for 20 min before being immunostained. *DAMPs staining*: Sections were immunostained with a goat anti-S100A9 polyclonal antibody (AF2065) by incubation with HISTOFINE simple stain mouse MAX-PO (G) for 30 min.

**Quantification of DAMPs in kidney**. DAMPs-DAB staining images were obtained and digitally captured using slide scanner VS200 (OLYMPUS, Germany) with a 20× objective lens. Digital image analysis was performed with commercial software HALO (IndicaLabs, Corrales, NM, USA). DAB and hematoxylin signals were detected using object colocalization-based algorithms, and the amount of extracellular DAMPs were estimated by subtracting the hematoxylin-colocalizing DAB signals from the total DAB signals.

**Flowcytometric analysis**. mProx24 cells, which are renal proximal tubule epithelial cells, were cultured on a 24-well plate in the presence of FITC-labeled crystals with/without AIM (100 μg/mL) in DMEM/F12 supplemented with 10% FBS for 1 h at 37 C. Thereafter, cells were harvested in 4 mL round-bottomed tubes. Fluorescent intensity for FITC was analyzed by flowcytometry (BD FACSCelesta, BD Biosciences). BD FACSDiva and FlowJo (BD Biosciences) were used for analysis.

**In vitro CaOx crystal development**. Crystals were precipitated by slowly mixing calcium chloride and sodium oxalate at final concentration of 1 and 5 mM, respectively, and each protein solution for 1 h. Crystals were collected by centrifugation, and observed by Polarized light optical microphotography (IX83, Olympus).

**Pull-down assay**. The amount of protein attached to crystals was evaluated by Western Blotting. Specifically, crystals developed by mixing calcium chloride and sodium oxalate were collected by centrifuge, and then incubated with either recombinant whole FLAG-tagged AIM protein (1 μg/mL) or each FLAG-tagged SRCR domain (0.3 μg/mL, which is at the identical molar conc. with that of whole AIM) at 37 °C for 1 h. Crystals were then centrifuged, washed by PBS twice, and the crystal pellets were boiled in SDS-loading buffer. The liquid phase was immunoblotted for AIM.

**Analysis of association and dissociation of negatively charged substances to crystals**. Crystals were incubated with rAIM (13.2 pmol) and each of OPN, $pAA_{5.1}$ or $D_9E_3$ peptide at different molar rations at 37 °C for 1 h. Thereafter, crystals were centrifuged, washed by PBS twice, and the amount of attached rAIM was analyzed by immunoblotting. To assess the dissociation of rAIM from crystals by the presence of other negatively charged substance, crystals were first incubated with rAIM (13.2 pmol) at 37 °C for 1 h. After being washed with PBS twice, the crystals were incubated with either of OPN, $pAA_{5.1}$ or $D_9E_3$ peptide at different molar rations for another hour. The amount of rAIM remaining at the crystals was assessed as above.

**Quantification of food intake**. Mice were maintained in metabolic cages (SN-783 No. 4B, Natsume Seisakusho Co., Ltd.) individually and the body weight and food intake were measured daily.

**Quantitative PCR assay**. The quantitative evaluation of mRNA was performed by the $\Delta\Delta C_T$ method using a QuantStudio 3 Real-Time PCR system (Thermo Fisher Scientific). Sequences of the oligonucleotides used are listed in Supplementary Table 1.

**Statistics and reproducibility**. Data were analyzed using BellCurve for Excel (Social Survey Research Information Co., Ltd.) and SAS ® 9.4 (SAS Institute Inc.) and are presented as mean values ± s.d. unless specified otherwise. Paired results were assessed using parametric tests such as Welch's *t*-test. For comparisons between multiple groups to compare each group against the control group, one-way, multi-way, or repeated measures ANOVA followed by the Dunnett's post hoc test were performed. One-way ANOVA followed by the Bonferroni's post hoc test was used for comparison between values. *$P < 0.05$, **$P < 0.01$, and ***$P < 0.001$, unless specified. Information on the reproducibility of experiments, including the sample sizes and number of replicates and how replicates were defined in the legends of figures.

**Reporting summary**. Further information on research design is available in the Nature Research Reporting Summary linked to this article.

## Data availability

All data generated or analyzed during this study are included in this published article (and its Supplementary Information files).

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

## Acknowledgements

We thank Dr. K. Kurokawa for helpful advice and discussion; Y. Yoshikawa and R. Iijima for helpful assistance in mouse maintenance. This work was supported by AMED-LEAP (JP22gm0010006h), Japan Agency for Medical Research Development (to T.M.), JSPS KAKENHI grant numbers 20H03446 and 21H05122 (to S.A.), 19K16535 (to N.M.) and 20K16213 (to K.T.), and Takeda Science Foundation (to S.A.).

## Author contributions

K.M. carried out major experiments, N. Maehara performed animal experiments and, histologic analysis, A.E. contributed to the protein production. K. Taniguchi and A.H. performed immunohistochemistry, K.Y. and A.N. did ELISA analysis, N. Matsuhashi, Y.S., R.I., Y.I., and K. Tanabe collected human samples and contributed to their analysis. S.A. did flowcytometric analysis, designed experiments together with T.M. and organized the figures, T.M. supervised the whole study and wrote the paper.

## Competing interests

The authors declare no competing interests.
