## [Peer Review File · Communications Biology]

Reviewers' comments:

Reviewer #1 (Remarks to the Author):

In the submitted manuscript the authors showed in a convincing way including in vivo experiments that the AIM protein significantly suppresses kidney stone development and improves stone-associated damage. This potentially has a huge therapeutic potential.

Minor comments:

1. In the Introduction (line 50) you wrote that global warming is associated with kidney stone formation. Could you provide any reference for this? What could be the mechanism behind this association?
- 2 line 97: 'disease animals', 'in' is missing
- 3.lines 299-301: the fact that epithelial cells surrounding the deposited stones were positive for KIM-1 does not prove causality, but association only.
4. purification of rAIM - could you please provide some data on the purity of the obtained protein? Also, was it FLAG-tagged? In Figure 2 C you wrote that in the pull-down experiment it was detected with anti-FLAG antibodies. Do you have any explanation why on this WB there are two bands for AIM in the pull-down fraction but not in the input? If the protein was FLAG-tagged how can you be sure that the native one would have the same effects? Please discuss.
5. line 535: 'otherwise' is missing at the end of the sentence.
6. for analysis of body weight changes and food intake repeated measure ANOVA should be use instead of multi-way ANOVA.
7. Legend to Figure 2 C: you wrote that you used 1 µg/ml for rAIM and 0.3 µg/ml for the domains, but in the methods section stays 100 µg/ml and 30 µg/ml, respectively. Please double-check.
8. Legend to Figure 6 - the font size is smaller that for the other legends. Please adjust.
9. how was the food intake quantified? Did you house the mice individually?
10. Figure 3 B-D: you wrote in that 3 mice in the control group died. Then why are there 4 data points for the control group on the graphs and not 1?
11. Why there is no quantification provided for the data on Figure 4?
12. Figure 5 - the intensity of the staining is very low. Have you tried to obtain photos of better quality? in the legend stays that the n number was 6, but on the graph the number of data points differ between groups. Were there some outlier that you excluded from the analysis?
13. Figure 6 D and E - should the kidney stone area be not the same for the day 6 (D) and KIM1 KO PBS group? why is it app. 8% in the first case and app. 5.5% in the second one?

Reviewer #2 (Remarks to the Author):

The study based on the inhibition of calcium oxalate crystal aggregation and adhesion, and the phagocytosis and digestion of calcium oxalate crystal by renal tubular epithelial cells, and proposed two distinct mechanisms of AIM and KIM-1 in the treatment of CaOx kidney stones: preventing the formation of stones and contributing to the removal of developed stones.

However, the concerns of some aspects are as follows:

1. As shown in line 98 of the manuscript, the author claimed the pathological similarity between kidney stone disease and AKI. Is there any reference to support this conclusion? As we know, in clinical situation, the development of kidney stone disease is mostly a chronic process, which is not as acute as the animal model with glyoxylate loading.
2. As shown is line 154 of the manuscript, the results here seems to be unconvincing. Is it reasonable to conclude that the decrease of injury and inflammation in the kidney is because of the alteration of the shape of the CaOx crystals?
3. As shown in line 266 of the manuscript. What's the role of DAMPs in the development of CaOx stones. Although AIM reduces DAMPs, the author did not explain why this mechanism is crucial in the

prevention of CaOx stone formation. It would be preferable for the author to present that the decrease of DAMPs by other methods can also prevent the CaOx deposition, so to affirm the importance of DAMPs and AIM in the prevention of CaOx formation.

4. This research actually presented two important proteins, AIM and KIM-1. However, the title of this paper seems to ignore KIM-1, and the relationship between them is not clear. As shown in line 135, the author concluded that the effect of rAIM was independent from the AIM/KIM-1 axis. So, what's the emphasis of this paper? AIM or KIM-1?

5. In the figure 1, 2 and 3, mRNA expression of inflammatory genes and injury genes was presented, however protein level should still be provided.

6. Overall, the clear presented mechanisms of AIM are only from two aspects. One is the disturbance of the aggregation and growth of CaOx crystals, another is the decrease of DAMPs. And for KIM-1, there is only an observation of elimination of developed crystals. The author spending a lot efforts to rule out some mechanisms of AIM or KIM-1, rather than further elucidate the actual mechanism of AIM or KIM-1 in the treatment of CaOx stones.

Responses to Reviewers' Comments

We thank the referees for their insightful and constructive comments during the initial review of our manuscript. We have revised the manuscript extensively in response to the concerns raised by the Reviewer as follows (the reviewer's comments are shown in blue).

We hope that our changes will satisfy the Reviewers.

To Reviewer 1:

(Minor comments only)

Q1. In the Introduction (line 50) you wrote that global warming is associated with kidney stone formation. Could you provide any reference for this? What could be the mechanism behind this association?

A1. A climate-related increase in the prevalence of kidney stone disease was mentioned in "Climate-related increase in the prevalence of urolithiasis in the United States" by Brikowski TH, Lotan Y, Pearle MS. Proc Natl Acad Sci U S A. 105, 9841-9816 (2008) (Our reference #4). This paper was also cited in "Kidney stones" by Khan, S. R. et al. Nat. Rev. Dis. Primers. 2, 16008 (2016) (Our reference #1). We modified the beginning sentence of Introduction to make the relation of references with the description clearer.

Q2. Line 97: 'disease animals', 'in' is missing

A2. Thank you for the note. This was corrected adequately.

Q3.lines 299-301: the fact that epithelial cells surrounding the deposited stones were positive for KIM-1 does not prove causality, but association only.

A3. We agree with the reviewer's indication, and thus, we modified the description according to the comment. However, KIM-1 is specifically expressed in the proximal tubular epithelial cells upon injury. Therefore, we believe that KIM-1 is expressed (positive) in the stone-surrounding epithelial cells which must be under physical stress or injury caused by the stone.

Q4. Purification of rAIM – could you please provide some data on the purity of the obtained protein? Also, was it FLAG-tagged? In Figure 2 C you wrote that in the pull-

down experiment it was detected with anti-FLAG antibodies. Do you have any explanation why on this WB there are two bands for AIM in the pull-down fraction but not in the input? If the protein was FLAG-tagged how can you be sure that the native one would have the same effects? Please discuss.

A4. In response to the concern, we showed the SDS-PAGE photo of the rAIM used in the animal experiments in Supplementary Figure 1a. The rAIM used in the experiments to address its therapeutic effect in mice (Fig. 1) was non-tagged native one, whereas the AIM and SRCR domains used in the pull-down experiment (Figure 2c) were FLAG-tagged. The same FLAG-tagged recombinant proteins were used for in vitro and in vivo assays to compare the functional efficiency in the full-size AIM and SRCR domains (Fig. 2d & Fig. 2e). We also showed the SDS-PAGE photo of the FLAG-tagged rAIM in Supplementary Figure 1a and the FLAG-tagged rSRCR proteins in Supplementary Figure 1b. In addition, we modified the Fig. 2c and described the details in the legend to avoid confusion.

In Figure 2c, the white line in the middle of the band of pull-downed AIM is a halation due to too high intensity of the signal and does not represent two bands. This was also additionally commented in the legend.

The multiple bands observed in both input and pull-down of SRCR1 and SRCR 2 are because of the glycosylation variation, which we always observe during the production of SRCR domains of AIM. This was also mentioned in the legend for Fig. 2c.

As the binding of the non-tagged full-size AIM to the crystals was obvious by the analysis using flowcytometer (Figure 2b), we are confident that the native AIM and SRCR domains have similar binding efficiencies to the crystals as those of FLAG-tagged ones. Unfortunately, because appropriate antibodies against each SRCR domain are not available, it is not possible to perform the pull-down assay using native SRCR domains.

Q5. line 535: 'otherwise' is missing at the end of the sentence.

A5. Thank you for the notice. We corrected adequately.

Q6. for analysis of body weight changes and food intake repeated measure ANOVA should be use instead of multi-way ANOVA.

A6. We re-analyzed the results by ANOVA and replaced the Figure 2h, 2i, 3e, and 3f.

Q7. Legend to Figure 2 C: you wrote that you used 1 µg/ml for rAIM and 0.3 µg/ml for the domains, but in the methods section stays 100 µg/ml and 30 µg/ml, respectively. Please double-check.

A7. “1 µg/ml for rAIM and 0.3 µg/ml for the domains” is correct. We corrected the description in the Methods section.

Q8. Legend to Figure 6 - the font size is smaller than for the other legends. Please adjust.

A8. We modified it adequately.

Q9. how was the food intake quantified? Did you house the mice individually?

A9. We added the detail of food intake quantification in the Methods section.

Q10. Figure 3 B-D: you wrote in that 3 mice in the control group died. Then why are there 4 data points for the control group on the graphs and not 1?

A10. As described in the legends, 3 out of 7 control or R8K4 peptide-injected mice died of kidney failure before day 6. Therefore, surviving 4 mice for each could be analyzed. Six mice were injected with either rAIM, OPN, pAA or D9E3 peptide, and all mice were survived at day 6.

Q11. Why there is no quantification provided for the data on Figure 4?

A11. Data of relative quantification are presented in the graphs. To make it clearer, we added “no competitors” instead of “Molecular ratio 0 vs AIM”.

Q12. Figure 5 - the intensity of the staining is very low. Have you tried to obtain photos of better quality? in the legend stays that the n number was 6, but on the graph the number of data points differs between groups. Were there some outlier that you excluded from the analysis?

A12. We appreciate the reviewer’s indication, but it is likely that our results are at the standard levels of IHC for S100A9, when compared to same histologic data in other reports. We are afraid that although we have tried to improve the figure, it was difficult to increase the quality of photos.

Regarding the numbers, we analyzed 7 control mice and 6 mice for other groups. In pAA5.1 injected mice, two circles are merged, and thus, one data point appears to be missing. We mentioned this in the legend.

Q13. Figure 6 D and E - should the kidney stone area be not the same for the day 6 (D) and KIM1 KO PBS group? why is it app. 8% in the first case and app. 5.5% in the second one?

A13. The amount of kidney stones varied between experiments due to the difference in lots of glyoxylate. Note however that we use the same lot in one experiment.

To Reviewer 2:

Q1. As shown in line 98 of the manuscript, the author claimed the pathological similarity between kidney stone disease and AKI. Is there any reference to support this conclusion? As we know, in clinical situation, the development of kidney stone disease is mostly a chronic process, which is not as acute as the animal model with glyoxylate loading.

A1. As the reviewer indicated, the clinical course is indeed not similar in two diseases (e.g. acute vs. chronic as the reviewer commented). Our description (line 98) means that both diseases are similar in a pathological aspect. In AKI, one of the most important initial events is the accumulation of dead cell debris in the lumen of the proximal renal tubules, which causes tubular inflammation and further tubular injury, leading to glomerular dysfunction. We thought that the situation of kidney stone disease is similar to that in AKI, namely, development and accumulation of intraluminal crystals also promotes inflammation and tubular injury, sometimes causing AKI due to the multiple tubular obstruction. Therefore, as AIM binds to the intraluminal debris and promotes their phagocytic removal thereby suppressing the disease progression, we hypothesized that AIM might induce the removal of crystals/stones and contribute to the suppression of kidney stone disease, like in AKI.

Q2. As shown in line 154 of the manuscript, the results here seem to be unconvincing. Is it reasonable to conclude that the decrease of injury and inflammation in the kidney is because of the alteration of the shape of the CaOx crystals?

A2. As demonstrated in Figure 1c, in rAIM-injected mice, inflammation was also decreased as evident by the reduction of inflammatory cytokine mRNA levels. However, we agree that we have no direct evidence proving that the decrease of spiny shape reduces the tubular injury that causes inflammation. We deleted the sentence.

Q3. As shown in line 266 of the manuscript. What's the role of DAMPs in the development of CaOx stones. Although AIM reduces DAMPs, the author did not explain why this mechanism is crucial in the prevention of CaOx stone formation. It would be preferable for the author to present that the decrease of DAMPs by other methods can also prevent the CaOx deposition, so to affirm the importance of DAMPs and AIM in the prevention of CaOx formation.

A3. We apologize that our description was not convincing sufficiently. Here we showed that DAMPs removal was beneficial in reduction of inflammation locally and systemically, which resulted in the improvement not only renal function but also general health states. We did not emphasize that decrease of DAMPs contributed to the prevention of CaOx crystal deposition. We modified the description to avoid misunderstanding.

Q4. This research actually presented two important proteins, AIM and KIM-1. However, the title of this paper seems to ignore KIM-1, and the relationship between them is not clear. As shown in line 135, the author concluded that the effect of rAIM was independent from the AIM/KIM-1 axis. So, what's the emphasis of this paper? AIM or KIM-1?

A4. Thank you for the comment. In this study, we unexpectedly found that AIM and KIM-1, which cooperates in the repair of AKI, independently contributed to the reduction of kidney stones; namely, AIM suppresses crystal aggregation and crystal growth to stone, whereas KIM-1 is likely to reduce "developed" stones. We are confident that both are important findings, and thus, along the reviewer's comment, we changed the title to introduce both findings.

Q5. In the figure 1, 2 and 3, mRNA expression of inflammatory genes and injury genes was presented, however protein level should still be provided.

A5. In response to this request, we attempted to quantify inflammatory cytokines in the kidney by ELISA using the kidney lysates. However, after many trials using various

ELISA kits, we found that ELISA did not work precisely with kidney tissue lysates, and thus, unfortunately, we could not quantify the inflammatory proteins in the kidney. In mouse sera, only IL-6 could be detected by ELISA, and the result showed a similar profile with that of QPCR (Supplementary Figs. 4a and 6a). In addition, we newly showed the immunohistochemistry for KIM-1 in the kidney (Supplementary Figs. 4b and 6b), which was also parallel with the QPCR results. We would like to note that in many publications, only QPCR is employed to show the inflammatory and injury states in tissues including kidney. Therefore, we are confident that our QPCR results and the newly obtained protein results (though they are partial) are sufficient to demonstrate the inflammatory and injury states in the kidney. We hope that the reviewer generously understand our efforts.

Q6. Overall, the clear presented mechanisms of AIM are only from two aspects. One is the disturbance of the aggregation and growth of CaOx crystals, another is the decrease of DAMPs. And for KIM-1, there is only an observation of elimination of developed crystals. The author spending a lot efforts to rule out some mechanisms of AIM or KIM-1, rather than further elucidate the actual mechanism of AIM or KIM-1 in the treatment of CaOx stones.

A6. We appreciate the reviewer's indication. In this study, we identified a new mechanism of how AIM inhibits crystal aggregation, which thereby prevents the CaOx stone development in mice. In addition, we demonstrated that the removal of DAMPs by AIM decreases the inflammation of kidney, contributing to the improvement of renal function and the general health state. On the other hand, we found that the presence of KIM-1 contributes to the removal of developed stones in an independent fashion of AIM. Although we are confident that this finding is novel and has an impact, however, as the reviewer commented, precise mechanism of how KIM-1 decreases the stones is still unclear. We emphasized the requirement of further studies to clarify the mechanisms of how KIM-1 is involved in the stone removal.

REVIEWERS' COMMENTS:

Reviewer #1 (Remarks to the Author):

All comments are addressed

Reviewer #2 (Remarks to the Author):

This manuscript version is now well organized and the major flaws have been solved. The author discovered novel modes of kidney stone suppression by two important proteins, which is helping us further explore the mechanism of protein-crystal interaction in the future. However, It's a pity that the protein level of inflammatory cytokines can not be detected using ELISA. Since QPCR is working while ELISA is not, it would be better if the protein level of inflammatory cytokines could be detected by western blot.

Responses to Reviewers' Comments

We thank the referees for their review of our revised manuscript. We would like to respond below to the single request by the Reviewer 2 (the reviewer's comment is shown in blue). We hope that our response could satisfy the Reviewer.

To Reviewer 2:

This manuscript version is now well organized and the major flaws have been solved. The author discovered novel modes of kidney stone suppression by two important proteins, which is helping us further explore the mechanism of protein-crystal interaction in the future. However, It's a pity that the protein level of inflammatory cytokines can not be detected using ELISA. Since QPCR is working while ELISA is not, it would be better if the protein level of inflammatory cytokines could be detected by western blot.

Response:

In response to the request, we performed western blot analysis for the cytokines using the lysates of kidney. We employed the lysate of macrophage cell lines (RAW264.7 and J774.1) after a stimulation by LPS in culture (300 ng/mL, 9 hours). As shown in the attached figure (next page), unfortunately, no obvious signals of any cytokine were detected in the kidney, either in the presence or absence of rAIM treatment. This is not due to technical problems of western blotting as the positive control provided signals.